# Effect of Binge-Drinking on Quality of Life in the ‘Seguimiento Universidad de Navarra’ (SUN) Cohort

**DOI:** 10.3390/nu15051072

**Published:** 2023-02-21

**Authors:** Rafael Perez-Araluce, Maira Bes-Rastrollo, Miguel Ángel Martínez-González, Estefanía Toledo, Miguel Ruiz-Canela, María Barbería-Latasa, Alfredo Gea

**Affiliations:** 1Department of Preventive Medicine and Public Health, University of Navarra, 31008 Pamplona, Spain; 2Instituto de Investigación Sanitaria de Navarra (IdiSNA), Navarra Institute for Health Research, 31008 Pamplona, Spain; 3Biomedical Research Network Center for Pathophysiology of Obesity and Nutrition (CIBEROBN), Carlos III Health Institute, 28029 Madrid, Spain

**Keywords:** alcohol, binge-drinking, quality of life, cohort study

## Abstract

Background: Binge-drinking is one of the alcohol drinking patterns with the worst health consequences. Nonetheless, binge-drinking is highly prevalent. The perceived benefits that motivate it are ultimately related to subjective well-being. In this context, we analyzed the relationship between binge-drinking and quality of life. Methods: We evaluated 8992 participants of the SUN cohort. We classified as binge-drinkers those who reported consuming six or more drinks on at least one occasion the year before recruitment (*n* = 3075). We fitted multivariable logistic regression models to calculate the odds ratios (ORs) of a worse physical and mental quality of life, measured with the validated SF-36 questionnaire at 8 years of follow-up (cut-off point = P_75_ or highest score). Results: Binge-drinking was associated with greater odds of having a worse mental quality of life, even adjusting for quality of life at 4 years of follow-up, used as an approximation to a baseline measure (OR = 1.22 (1.07–1.38)). This value was mainly due to the effects on vitality (OR = 1.17 (1.01–1.34)) and mental health (OR = 1.22 (1.07–1.39)). Conclusions: Binge-drinking may lead to poorer mental quality of life; therefore, binge-drinking for enhancement purposes does not seem to be justified by this effect.

## 1. Introduction

According to the latest report of the World Health Organization (WHO), alcohol consumption leads to about 3 million deaths per year and is also the third leading lifestyle-related risk factor with the highest burden of disease in the world [1,2]. This is related to two combined factors, high consumption around the world, both in prevalence and quantity of alcohol consumed, and the cause–effect relationship of alcohol with multiple diseases and injuries [2].

Alcohol can be associated, among other diseases, with different types of cancer, cardiovascular diseases, diabetes, liver diseases and mental illness, but also with an increase in interpersonal violence, suicides and unintentional accidents [2,3]. However, despite having so many adverse effects, alcohol is consumed by more than half of the adult population (>15 years old) in many regions of the world, with Europe being the one with the highest consumption, with an average alcohol intake among drinkers of 37.4 g/day [2]. Moreover, much of this consumption is made in the context of binge-drinking, one of the consumption patterns with the worst consequences for health, not only in the short term but also in the long term [4,5], which entails both health and large economic costs [6].

Binge-drinking is most commonly defined as the consumption of four alcoholic drinks for females and five for males on the same occasion [7].

Binge-drinking can be psychologically explained by its subjective benefits in the short term, which are ultimately related to subjective well-being. Enhancement is the most common motive claimed by binge-drinkers, while coping motives are associated with alcohol-related problems, and social motives are also typical for moderate drinkers [8,9]. In Spain, the main reasons given for alcohol consumption are that consumers like the way they feel after drinking and that it is fun and encourages partying [10]. Effects with which excessive alcohol consumption is most often associated in advertisements and media, having these platforms play an important role as remote determinants of this behavior [11,12]. This, repeated over time, should lead, at least from their point of view, to better subjective well-being. One approach to measuring this effect over time could be with the mental quality of life, which is measured with a validated questionnaire, the SF-36 [13], which measures several domains of quality of life from which summary measures of physical quality of life and mental quality of life can also be calculated. It is precisely in mental quality of life where previous studies have observed significant associations [14,15,16,17,18,19,20]. This effect could be supported by the consequences of binge-drinking on some domains of mental quality of life that have already been described, such as socialization [21] or mental health [22], as binge-drinking has been associated with an increased risk of depression [23,24]. However, this effect has also been observed in reverse [22,25], so it is impossible to establish causality in cross-sectional studies, and many of those published so far have that design.

In addition, an important factor to consider when studying the effects of binge-drinking is an adjustment for the amount of alcohol consumed in order to not confound the effect of alcohol per se with the effect of alcohol consumption on binge-drinking. Most previous studies have not taken this variable into account.

Moreover, previous articles have suggested that binge-drinking does not affect men and women in the same way [15,16,26,27,28].

In this article, we aimed to assess the prospective association between binge-drinking and subsequent quality of life in alcohol drinkers from a Mediterranean cohort and to explore a potential modification of this association by sex.

## 2. Materials and Methods

### 2.1. Study Sample

The SUN (‘Seguimiento Universidad de Navarra’) Project is a prospective and multipurpose cohort study designed to evaluate different aspects of dietary patterns and lifestyles, relating them to health outcomes. By the end of 2019, it already had almost 23,000 participants, who were assessed every 2 years using self-administered questionnaires. It is a dynamic cohort, so although recruitment began in 1999, it is permanently open. Participants are university graduates from all over Spain. The methods and many specific details of the SUN cohort have been previously described [29,30].

As the outcome is measured at 8 years of follow-up, from the initial sample of 22,894 participants, we excluded those who died before completing the 8-year follow-up questionnaire (*n* = 238) and those with less than 8 years in the cohort (*n* = 1481). From the remaining 21,175, we excluded 1518 lost to follow-up participants (retention rate: 92.8%). Those subjects who did not complete both SF-36 questionnaires at 4 and 8 years of follow-up were excluded (*n* = 6393 and *n* = 593, respectively) as well as those with insufficient data (*n* = 593). We also excluded 1059 participants with comorbidities that should have led them to stop drinking alcohol (heart attack, stroke, cancer, gastric ulcer and atrial fibrillation) and 2391 abstainers to avoid the potential sick-quitter effect and to compare the drinking pattern (binge-drinking or not) only among drinkers. We obtained a final sample of 8992 participants for our main analysis (Figure 1).

### 2.2. Outcome Measurement

We used the validated Spanish version of the 36-Item Short Form Health Survey (SF-36) to measure the quality of life. This questionnaire has been widely used in the scientific literature [31,32]. It contains 36 items, whose answers are coded, summed and transformed to a 0–100 scale in which 0 is the worst score and 100 is the best, corresponding to the lowest and highest quality of life, respectively. The transformation was made according to the valid and reliable method described by Vilagut et al. [13], which takes into account the mean values for the Spanish population. We also calculated the aggregated physical and mental components of the quality of life.

In our cohort, the SF-36 questionnaire was assessed after 4 and 8 years of follow-up. The measure at 8 years will be our outcome, and the measure at 4 years will be used as an approximation to a baseline measure in some of our adjustment models.

We decided to dichotomize the scores of the different dimensions since they were very skewed, and our interest was also in achieving a higher quality of life. Participants who were above the 75th percentile or who had the maximum value were classified as having a better quality of life, and the rest were classified as having a worse quality of life. We gave a value of 0 to those with a better quality of life and a value of 1 to those with a poorer quality of life. Different cut-off points were used for men and women.

In an additional analysis, we used the variables in a continuous form to establish the adjusted mean of each quality of life dimension according to whether or not they were binge-drinking.

### 2.3. Binge-Drinking Assessment

In the SUN project, the baseline questionnaire collects information on the maximum number of alcoholic beverages consumed by the participants on a weekday, on a weekend day and on celebrations and special occasions in the year prior to completing the questionnaire. This variable is collected categorically according to the following ranges: none, 1–2 drinks, 3–5 drinks, 6–9 drinks, 10–14 drinks and ≥15 drinks. Thus, for our classification, we included in the binge-drinking group those who consumed 6 or more drinks in a day. This definition implies a limitation, given the way in which our variable was collected, and is discussed below.

### 2.4. Other Covariates

At baseline, questionnaires were used for gathering self-reported information on demographic characteristics (age, sex, years of university education, marital status, job occupation and profession), lifestyle habits (smoking status, alcohol intake, sleeping hours and physical activity), personality scores (competitiveness, anxiety and phycological dependence), and anthropometric and clinical data (weight, height, food consumption, drug use and comorbidities). The self-reported weight and height values, as well as the physical activity and food consumption frequency questionnaires, have been previously validated [33,34,35,36,37].

From these variables, we calculated the body mass index (BMI), the total energy intake and the adherence to the Mediterranean diet with the Mediterranean Dietary Score (MDS) [38].

We also included the incidence in the multivariable adjustment, during the follow-up, of some diseases that could have an important effect on the quality of life, such as cancer, depression, diabetes or cardiovascular disease.

### 2.5. Statistical Analysis

Baseline variables are described according to the binge-drinking habit and adjusted for age and sex with the inverse probability weighting method. For categorical variables, we calculated the percentage of participants included in each group, and for quantitative variables, we calculated the mean and standard deviation or the median and the interquartile range if the distribution of the variable did not follow a normal distribution.

To assess the possible effect of binge-drinking on the different dimensions of quality of life, we used multivariable logistic regressions. The odds ratios obtained from these analyses are of not being in the group with the highest quality of life, that is, of belonging to the first 3 quartiles. Thus, odds ratios with values greater than 1 indicate a detrimental effect, and odds ratios below 1 indicate a beneficial effect. For every odds ratio, their 95% confidence intervals (CI) were calculated. The multivariable logistic regression model included the following variables: age; years of university studies; marital status; job occupation; being a health professional; physical activity; sleeping hours; alcohol intake; smoking status; total energy intake; adherence to the MDS; body mass index (BMI); polypharmacy (self-reported consumption of 3 or more medications); prevalent diseases (diabetes, hypertension, hypercholesterolemia, arthritis, depression or pulmonary disease); and incidence of cancer, depression, diabetes or cardiovascular disease during the follow-up. In an additional model, we also adjusted for the quality of life after a 4-year follow-up as an approximation to a baseline measure (model 2 of adjustment).

We also performed stratified analyses according to sex, comorbidities and age (above or below 50 years). Furthermore, as sensitivity analyses, we repeated the analysis, including those participants who met the exclusion criteria for comorbidities and alcohol abstinence and also excluding those who only reported binge-drinking on special occasions.

Finally, we included the analysis using the dimensions of quality of life as continuous. By using multiple linear regressions, we obtained the adjusted mean value of each dimension of quality of life for each group according to their binge-drinking habit.

STATA 16 software (StataCorp. 2019. Stata Statistical Software: Release 16. College Station, TX, USA: StataCorp LLC.) was used for data analysis.

## 3. Results

Of the 8992 participants, 34.2% (*n* = 3075) were classified as binge-drinkers at baseline. This group had a lower proportion of women (44.9% in the binge-drinking group vs. 62.4% in the non-binge-drinking group) and was younger on average (with a mean age of 34.3 years (SD = 10.1) in the binge-drinking group vs. 40.2 (SD = 11.7) in the non-binge-drinking group). The main differences between groups after adjustment for age and sex (see Table 1) were related to total grams of alcohol per day and tobacco consumption, as expected, both higher in the binge-drinking group. In addition, a somewhat higher proportion of comorbidities was found in that group.

The incidence of diseases during the 8 years of follow-up was also of similar magnitude in both groups after adjustment for age and sex. In the non-binge-drinking group, we found an incidence of 1.7% of cancer, 0.3% of cardiovascular disease, 5% of depression and 1.0% of diabetes, while in the binge-drinking group, the data were 1.9%, 0.3%, 4.2% and 1.0%.

The results obtained for the physical items of quality of life are shown in Table 2. For the aggregated physical dimensions, the results are around unity, and for each item, they are mostly non-significant. We found a 13% relative increase (OR = 1.13 (1.02–1.26)) in the probability of presenting worse physical functioning, with the almost identical result after adjustment for quality of life after 4 years of follow-up. When stratifying by sex, the significance was maintained, with an increase in the magnitude of the effect, only in women, even with the adjustment of quality of life at a 4-year follow-up. However, *p* for interaction was not significant. The effect on bodily pain was around 14% (OR = 1.14 (1.03–1.27)), but it was not statistically significant after the adjustment of quality of life at 4-year follow-up (OR = 1.11 (1.00–1.24)).

Table 3 presents the association between binge-drinking and mental quality of life. In model 1, we found a statistically significant association for the aggregated mental dimensions, with a relative increase in the odds of belonging to the lowest rating categories of 27% (OR = 1.27 (1.13–1.43)), 27% for vitality (OR = 1.27 (1.14–1.42)) and 29% for mental health (OR = 1.29 (1.14–1.47)). When stratifying by sex, the results were similar for the aggregated mental dimensions. Regarding vitality and mental health, a greater effect was observed in men (36 vs. 18% in vitality and 30 vs. 22% in mental health), although the *p* for interaction was only significant for mental health. In model 2, a statistically significant association was maintained in these three items with a relative increase in the odds of obtaining a worse score for the aggregated mental dimensions (OR = 1.22 (1.07–1.38)), vitality (OR = 1.17 (1.01–1.34) and mental health (OR = 1.22 (1.07–1.39)). However, when stratifying by sex, the statistically significant association in the aggregate measure remained significant only for women (OR = 1.22 (1.02–1.44)). No significant association was found between social functioning or emotional role and binge-drinking.

Figure 2 shows, graphically, the values of the multivariable-adjusted odds ratio of having worse results for each quality of life item according to binge-drinking. The results are after 8 years of follow-up and with a multivariable adjustment, which includes the 4 years of follow-up quality of life data (model 2 of adjustment). In this summary figure, we can see that binge-drinking was not associated with better quality of life in any category.

For none of the stratified analyses, a statistically significant *p* for interaction was found (see Table 4). However, it should be noted that in each analysis, a significant odds ratio for the aggregated mental dimensions was obtained in only one of the strata for those with comorbidities, participants over 50 years of age, and those with an alcohol consumption ≤ 5 g/day.

We performed a sensitivity analysis by including participants with comorbidities that should have led them to stop drinking alcohol (heart attack, stroke, cancer, gastric ulcer and atrial fibrillation) and/or abstainers at baseline (Table 5). When including each group, the statistically significant associations remained significant, and with a barely modified magnitude, only a small decrease in mental quality of life was observed in women if abstainers were included. When both groups were included, the associations lost some strength, and even the association between mental quality of life and binge-drinking in women for model 2 of adjustment lost its statistical significance. Again, the statistically significant associations remained significant when we excluded those who only reported binge-drinking consumption on special occasions.

Finally, in Table 6, we show the multivariable-adjusted mean (including the quality of life data at 4 years of follow-up in the adjustment) for each item of quality of life for binge-drinkers and non-binge-drinkers. For all items, the mean value was lower for the binge-drinking group, although the confidence intervals overlapped, and a statistically significant result was only obtained for mental health and social functioning.

## 4. Discussion

Our results show that binge-drinking is associated with worse mental quality of life, especially poorer vitality and mental health. Moreover, no positive effect was seen for any of the items. In fact, when the outcome was analyzed as a continuous variable, participants in the binge-drinking group had worse scores on all quality-of-life dimensions.

It is important to note that the mental components are, precisely, the ones that want to be enhanced with binge-drinking [8,9]. Effective communication of this message to the population could decrease the high prevalence of binge-drinking.

These conclusions are very similar to those reached in previous studies [14,15,16,17,18,19,20]. Those that also used the SF-36 (or the short version, the SF-12) found similar results, with a greater magnitude of the association for the mental quality of life and non-statistically significant results for the physical component [15,17,19]. The other studies used the number of physically or mentally unhealthy days and also found a bigger association for the mental component [14,18,20]. However, these studies, except for one [17], are of cross-sectional design, so the temporal sequence cannot be established. In addition, there are studies that support that a poorer quality of life leads to higher alcohol consumption in binge-drinking [22,25].

To evaluate the effect of consumption on binge-drinking, we used alcohol consumption (in grams per day) as an adjustment variable. Thus, the effect we observed is independent of the total amount consumed. This allows us to observe what the specific effect of the pattern is, differentiating those in the binge-drinking group who concentrate the alcohol consumption on a single occasion and those in the non-binge-drinking group who spread the consumption throughout the week.

Regarding the difference between sexes, our results are nonconclusive. Although the magnitude of the OR found is higher in men, in general, significant associations (including after adjustments) were found for the general population or for women. Moreover, no statistically significant *p* for interaction was obtained for any of the items. Future studies could explore this issue further.

Our stratified analyses suggest that the association between binge-drinking and worse quality of life is stronger in young people, those with comorbidities and those with lower mean alcohol consumption, but again the *p* for interaction obtained were non-statistically significant. This would be consistent with binge-drinking being more frequent in young people [2,21] and with alcohol aggravating comorbidities [4].

Regarding the biological mechanisms that could explain this association, binge-drinking leads to elevated blood alcohol concentrations. In addition, the peculiar solubility of alcohol (it is both fat-soluble and water-soluble) means that it can reach many organs, where it can cause damage either directly or through its metabolites formed in the liver [4]. Among the mechanisms involved are the formation of reactive oxygen species, with the consequent oxidative stress, adduct formation, cell membrane disruption, modulation of the immune response or mitochondrial damage, among others [4]. All this leads to the development of different diseases, such as cirrhosis, which have been studied in depth, and the mechanisms by which it occurs are very well described and studied. However, there are many others, especially at the digestive level, in first and direct contact with alcohol or even at the cardiovascular or musculoskeletal level. In addition, it also leads to the worsening of prevalent diseases. This would explain the effect found on some physical items of quality of life, and the greater effect for those with comorbidities in the stratified analysis, as high concentrations of alcohol could aggravate previous damage caused by these diseases [4].

The brain is particularly affected by alcohol consumption. As already mentioned, alcohol has peculiar solubility properties, which allow it to penetrate the blood–brain barrier [4]. At the brain level, high concentrations of alcohol can change the expression, affinity or receptors of different neurotransmitters (especially γ-aminobutyric acid (GABA)) and even produce structural changes with altered synapses, neurogenesis or gene expression [5]. These mechanisms could be the biological basis that explains the negative effect of binge-drinking on the quality of mental life since the reward mechanisms are altered; with the capacity for enjoyment, performance decreases and memory is affected. There are also studies that have established the effect of binge-drinking on causing depression [23,24]. Other explanations could be mediated by the effect on the social or professional spheres [21]. However, the effect on social life should be reflected in the social functioning dimension, in which no statistically significant results were found. The effect on the professional sphere could be found in the physical and mental roles.

Regarding the limitations of our study, it is possible that we are underestimating the effect because of the way in which binge-drinking was measured. The binge-drinking definition was limited by the way this habit was recorded in the questionnaires of the SUN Project. On the one hand, cut-off points are usually established for the consumption of more than four or five alcoholic beverages (in men and women, respectively) [39,40]. We could set the cut-off at four or six, so, in an attempt to be conservative, as mentioned, we set the cut-off point at six drinks. On the other hand, our definition was the maximum number in a day, a definition that has been used in several studies [40], although 2 h as the time interval is more accurate to be used [40]. In addition to this, the maximum number of drinks is in the last year. For all these reasons, it is very likely that the binge-drinking group included people who are not real binge-drinkers or who have practiced binge-drinking very infrequently. All of this would lead us to underestimate the effect of binge-drinking, especially for women, for whom the usual cut-off point is further away from the one used in this article. In spite of all this, it is precisely in women where we found the most statistically significant results.

In addition, our exposure is only collected at baseline, and the binge-drinking habit could vary during the 8-year of follow-up when the outcome is measured. This was the direction of the sensitivity analysis that excluded those who reported high alcohol consumption only on special occasions. These analyses showed a slight increase in the magnitude of the effect on the mental component, which reinforces the idea that the effect is underestimated. Additionally, binge-drinking peaks in early adulthood [2], so it is more likely that this habit decreases during the follow-up.

In terms of strengths, one is the large number of variables collected at the beginning of the study, many of which were even validated. This allows us to make a good adjustment in the statistical analysis to limit confounding effects. The design of the SUN project also implies the greatest strength of this article: being able to establish a temporal sequence. In model 1 of the analysis, we ensured the temporal sequence by measuring exposure (binge-drinking) at baseline and outcome (quality of life items) at 8 years of follow-up. In model 2, in addition, we included in the adjustment the values of these items at 4 years of follow-up as an approximation to a baseline measure. With this second adjustment, we lose the magnitude of the effect, but by adding reference values, we ensure that the results obtained are due to alcohol consumption in binge-drinking. However, the fact that the reference data were at 4 years of follow-up and not at baseline may, once again, lead us to underestimate the effect since the effect of binge-drinking on the quality of life for the first 4 years of follow-up is included in these baseline values.

## 5. Conclusions

Our results suggest that binge-drinking causes a poorer quality of mental life; therefore, binge-drinking for enhancement motives is not justified by its effects on quality of life. Regarding the difference between sexes, our results are nonconclusive. For future studies, it could be worthwhile to look more closely at the different effects on men and women.

## Figures and Tables

**Figure 1 nutrients-15-01072-f001:**
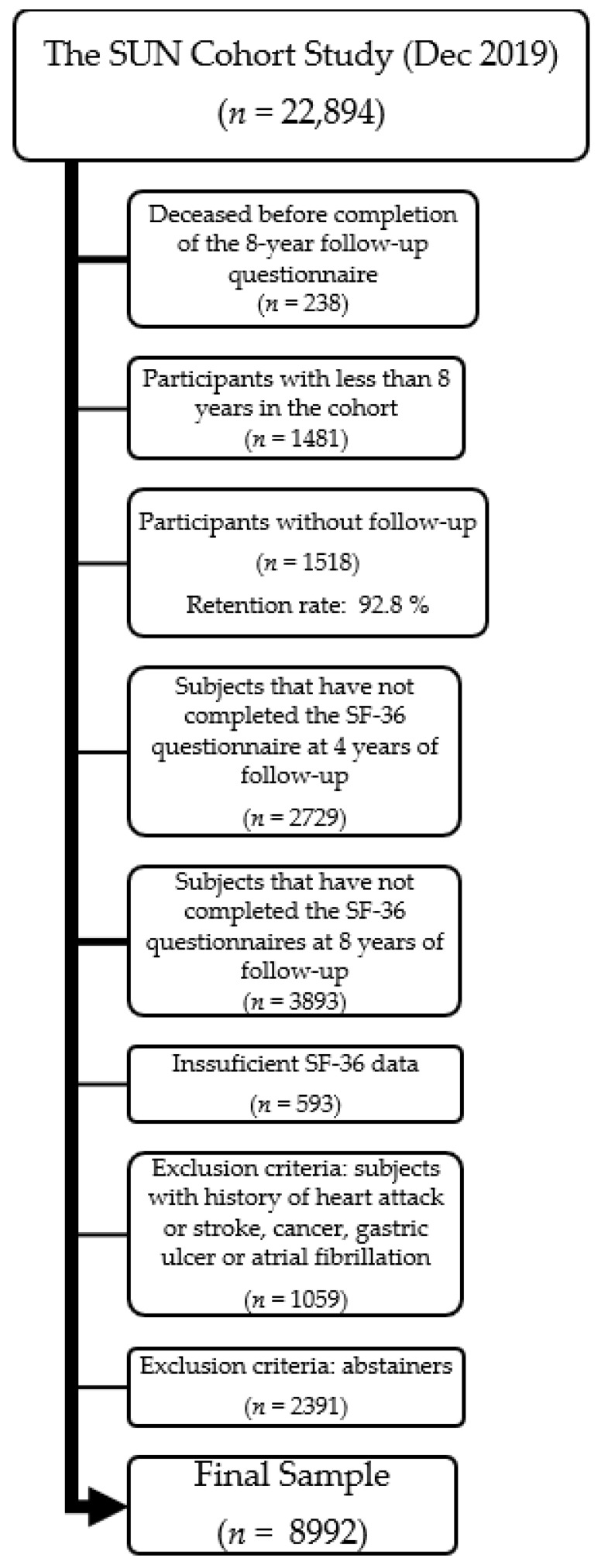
Flowchart of participants in the SUN Project: exclusion criteria and final sample.

**Figure 2 nutrients-15-01072-f002:**
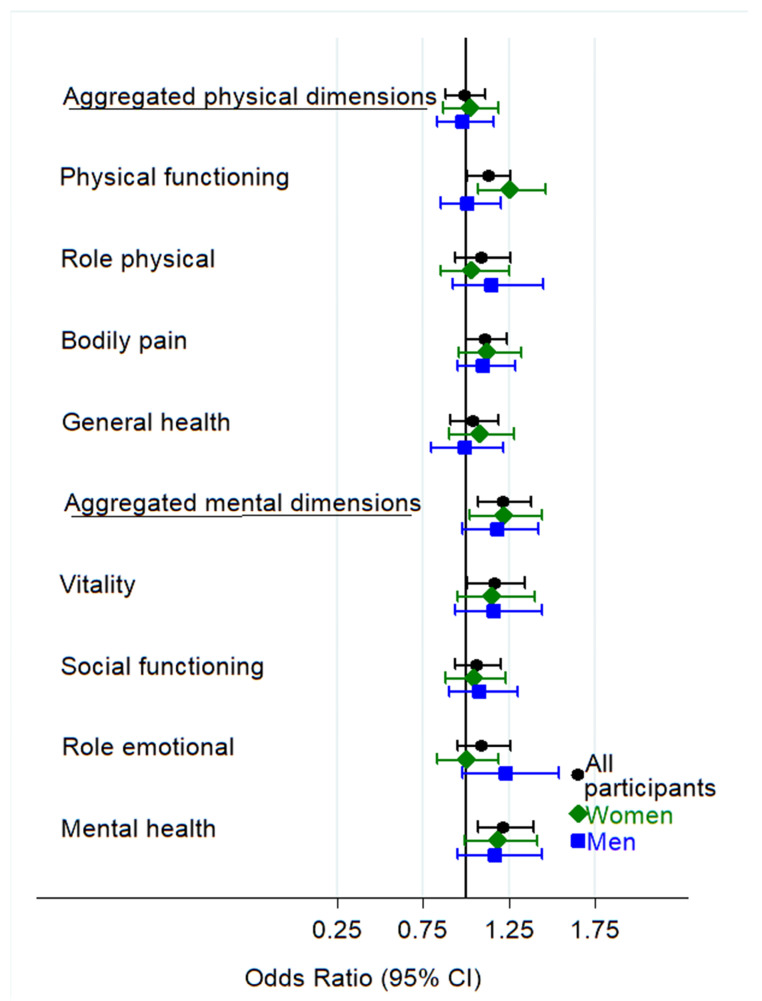
Multivariable adjusted * odds ratios of having worse results (<P_75_) for each quality of life item measured with the SF-36 according to binge-drinking (BD) habit after 8 years of follow-up, their 95% confidence intervals (CI) for the total sample and stratified by sex. * Adjusted for age; years of university studies; marital status; job occupation; being a health professional; physical activity; sleeping hours; alcohol intake; smoking status; total energy intake; adherence to the MDS; BMI; polypharmacy; prevalent diseases (diabetes, hypertension, hypercholesterolemia, arthritis, depression or pulmonary disease); incidence of cancer, depression, diabetes or cardiovascular disease, and for the 4 years of follow-up quality of life data.

**Table 1 nutrients-15-01072-t001:** Age and sex-adjusted baseline characteristics of the participants according to their binge-drinking habit. The SUN Project 1999–2019.

Basal Characteristics	No Binge-Drinking (*n*= 5917)	Binge-Drinking (*n* = 3075)
**Demographic characteristics**		
Years of university education, median (IQR)	5 (4–5)	5 (4–5)
Marital status (%)		
Single	43.1	42.9
Married	52.6	52.5
Other	4.3	4.6
Job occupation (%)		
Full time	84.2	84.9
Part time	8.5	7.2
Housekeeper	1.9	1.8
Unemployed	3.8	4.1
Retired	1.6	2.1
Health professionals (%)	54.1	53.6
**Lifestyle habits**		
Physical activity (METS-h/week), median (IQR)	16.5 (6.4–30.7)	16.8 (5.7–30.8)
Sleeping hours (h/day), mean (SD)	7.1 (0.9)	7.2 (0.9)
Alcohol intake (g/day), median (IQR)	3.5 (1.6–8.8)	8.1 (3.5–15.0)
Smoking status (%)		
Current smoker	19.3	28.3
Former smoker Lifetime tobacco exposure (pack-years), mean (SD)	28.0 4.9 (8.9)	36.7 7.8 (11.1)
**Anthropometric and clinical data**		
Total energy intake (kcal/d), median (IQR)	2434 (1993–2966)	2424 (1946–2986)
Adherence to the MDS * (0 to 8 score), mean (SD)	4.0 (1.7)	3.9 (1.7)
BMI (kg/m^2^), mean (SD)	23.3 (3.3)	23.9 (3.6)
Type 2-diabetes mellitus (%)	1.4	1.0
Hypertension (%)	9.4	12.8
Hypercholesterolemia (%)	16.5	18.1
Arthritis (%)	1.6	2.2
Depression (%)	10.2	10.1
Pulmonary disease (%)	6.6	7.6
Polypharmacy (%)	2.0	2.2
**Personality scores (range, 0 to 10), mean (SD)**		
Competitiveness	7.0 (1.7)	6.9 (1.8)
Anxiety	6.0 (2.2)	6.0 (2.2)
Phycological dependence	3.6 (2.8)	3.7 (2.9)

MDS: Mediterranean diet score. BMI: body mass index. * MDS excluding the item related to alcohol consumption.

**Table 2 nutrients-15-01072-t002:** Multivariable adjusted odds ratios of having worse quality of life on the physical dimensions of quality of life, measured with the SF-36 according to binge-drinking (BD) habit after 8 years of follow-up, their 95% confidence intervals (CI) for the total sample and stratified by sex.

		All Participants		Women		Men		
BD/No BD		3105/6037	*p*-Value	1391/3772	*p*-Value	1714/2265	*p*-Value	*p* for Interaction
		Multivariable-Adjusted OR (95% CI)		Multivariable-Adjusted OR (95% CI)		Multivariable-Adjusted OR (95% CI)		
**Aggregated physical dimensions**	Model 1	1.00 (0.89–1.12)	0.960	1.02 (0.88–1.20)	0.772	0.99 (0.84–1.17)	0.923	0.957
	Model 2	0.99 (0.88–1.11)	0.881	1.01 (0.86–1.19)	0.897	0.99 (0.83–1.17)	0.896	0.904
**Physical functioning**	Model 1	1.13 (1.02–1.26)	0.020	1.24 (1.07–1.43)	0.004	1.04 (0.89–1.21)	0.623	0.654
	Model 2	1.13 (1.00–1.26)	0.041	1.25 (1.07–1.46)	0.006	1.01 (0.85–1.20)	0.895	0.290
**Role physical**	Model 1	1.09 (0.95–1.26)	0.213	1.02 (0.85–1.24)	0.806	1.17 (0.93–1.47)	0.170	0.298
	Model 2	1.09 (0.94–1.26)	0.258	1.03 (0.85–1.25)	0.784	1.15 (0.92–1.45)	0.226	0.405
**Bodily pain**	Model 1	1.14 (1.03–1.27)	0.013	1.14 (0.98–1.33)	0.099	1.14 (0.99–1.32)	0.070	0.657
	Model 2	1.11 (1.00–1.24)	0.061	1.12 (0.95–1.31)	0.165	1.10 (0.95–1.29)	0.208	0.767
**General health**	Model 1	1.04 (0.92–1.17)	0.561	1.04 (0.88–1.22)	0.667	1.02 (0.84–1.23)	0.836	0.193
	Model 2	1.04 (0.91–1.19)	0.535	1.07 (0.90–1.28)	0.430	0.99 (0.80–1.21)	0.899	0.584

Model 1: Adjusted for age, years of university studies, marital status, job occupation, being a health professional, physical activity, sleeping hours, alcohol intake, smoking status, total energy intake, adherence to the MDS, BMI, polypharmacy, prevalent diseases (diabetes, hypertension, hypercholesterolemia, arthritis, depression or pulmonary disease) and incidence of cancer, depression, diabetes or cardiovascular disease. Model 2: Model 1 also adjusted for the 4 years of follow-up quality of life data.

**Table 3 nutrients-15-01072-t003:** Multivariable adjusted odds ratios of having worse quality of life on the mental dimensions of quality of life, measured with the SF-36 according to binge-drinking (BD) habit after 8 years of follow-up, their 95% confidence intervals (CI) for the total sample and stratified by sex.

		All Participants		Women		Men		*p* for Interaction
BD/No BD		3105/6037	*p*-Value	1391/3772	*p*-Value	1714/2265	*p*-Value	
		Multivariable-Adjusted OR (95% CI)		Multivariable-Adjusted OR (95% CI)		Multivariable-Adjusted OR (95% CI)		
**Aggregated mental dimensions**	Model 1	1.27 (1.12–1.43)	<0.001	1.25 (1.06–1.48)	0.008	1.24 (1.04–1.47)	0.015	0.223
	Model 2	1.21 (1.07–1.37)	0.002	1.22 (1.02–1.44)	0.026	1.18 (0.98–1.41)	0.084	0.375
**Vitality**	Model 1	1.27 (1.14–1.42)	<0.001	1.18 (1.01–1.37)	0.039	1.36 (1.16–1.60)	<0.001	0.094
	Model 2	1.17 (1.01–1.34)	0.031	1.15 (0.95–1.40)	0.141	1.16 (0.94–1.44)	0.156	0.799
**Social functioning**	Model 1	1.09 (0.98–1.23)	0.120	1.06 (0.91–1.23)	0.468	1.12 (0.94–1.33)	0.204	0.486
	Model 2	1.06 (0.94–1.20)	0.327	1.04 (0.88–1.23)	0.626	1.08 (0.90–1.30)	0.419	0.480
**Role emotional**	Model 1	1.11 (0.97–1.27)	0.142	0.99 (0.83–1.18)	0.938	1.27 (1.02–1.58)	0.036	0.138
	Model 2	1.09 (0.95–1.26)	0.206	0.99 (0.83–1.19)	0.948	1.23 (0.98–1.54)	0.073	0.223
**Mental health**	Model 1	1.29 (1.14–1.47)	<0.001	1.22 (1.03–1.45)	0.021	1.30 (1.07–1.58)	0.008	0.025
	Model 2	1.22 (1.07–1.39)	0.004	1.18 (0.99–1.41)	0.069	1.18 (0.96–1.45)	0.121	0.143

Model 1: Adjusted for age, years of university studies, marital status, job occupation, being a health professional, physical activity, sleeping hours, alcohol intake, smoking status, total energy intake, adherence to the MDS, BMI, polypharmacy, prevalent diseases (diabetes, hypertension, hypercholesterolemia, arthritis, depression or pulmonary disease) and incidence of cancer, depression, diabetes or cardiovascular disease. Model 2: Model 1 also adjusted for the 4 years of follow-up quality of life data.

**Table 4 nutrients-15-01072-t004:** Multivariable adjusted * odds ratios of having worse physical and mental quality of life according to binge-drinking (BD) habit after 8 years of follow-up and their 95% confidence intervals (CI) for different subgroups.

		Multivariable-Adjusted * OR (95% CI)	*p*-Value	*p* for Interaction
Participants with comorbidities (*n* = 3757)	Aggregated physical dimensions	1.11 (0.91–1.35)	0.297	0.219
	Aggregated mental dimensions	1.28 (1.05–1.56)	0.015	0.368
Participants without comorbidities (*n* = 5235)	Aggregated physical dimensions	0.93 (0.81–1.08)	0.359	0.219
	Aggregated mental dimensions	1.16 (0.99–1.37)	0.062	0.368
Age ≥ 50 (*n* = 1580)	Aggregated physical dimensions	1.05 (0.70–1.58)	0.821	0.641
	Aggregated mental dimensions	1.29 (0.93–1.79)	0.123	0.613
Age < 50 (*n* = 7412)	Aggregated physical dimensions	0.98 (0.87–1.10)	0.707	0.641
	Aggregated mental dimensions	1.21 (1.05–1.38)	0.007	0.613
Alcohol intake >5 g/day (*n* = 4448)	Aggregated physical dimensions	0.95 (0.81–1.12)	0.558	0.728
	Aggregated mental dimensions	1.17 (0.99–1.38)	0.067	0.786
Alcohol intake ≤5 g/day (*n* = 4544)	Aggregated physical dimensions	1.03 (0.87–1.23)	0.703	0.728
	Aggregated mental dimensions	1.27 (1.05–1.55)	0.016	0.786

* Adjusted for age, years of university studies, marital status, job occupation, being a health professional, physical activity, sleeping hours, alcohol intake, smoking status, total energy intake, adherence to the MDS, BMI, polypharmacy and prevalent diseases (diabetes, hypertension, hypercholesterolemia, arthritis, depression or pulmonary disease) and for the 4 years of follow-up quality of life data.

**Table 5 nutrients-15-01072-t005:** Multivariable adjusted Odds ratios of having worse physical and mental quality of life on the mental dimensions of quality of life, measured with the SF-36 according to binge-drinking (BD) habit after 8 years of follow-up, their 95% confidence intervals (CI) for the total sample and stratified by sex. Participants previously excluded according to exclusion criteria were included.

			All Participants Multivariable-Adjusted OR (95% CI)	*p*-Value	Women Multivariable-Adjusted OR (95% CI)	*p*-Value	Men Multivariable-Adjusted OR (95% CI)	*p*-Value
**Main analysis**	Aggregated physical dimensions	Model 1	1.00 (0.89–1.12)	0.960	1.02 (0.88–1.20)	0.772	0.99 (0.84–1.17)	0.923
		Model 2	0.99 (0.88–1.11)	0.881	1.01 (0.86–1.19)	0.897	0.99 (0.83–1.17)	0.896
	Aggregated mental dimensions	Model 1	1.27 (1.12–1.43)	<0.001	1.25 (1.06–1.48)	0.008	1.24 (1.04–1.47)	0.015
		Model 2	1.21 (1.07–1.37)	0.002	1.22 (1.02–1.44)	0.026	1.18 (0.98–1.41)	0.084
**With those with comorbidities**	Aggregated physical dimensions	Model 1	1.01 (0.91–1.13)	0.805	1.06 (0.91–1.23)	0.447	0.99 (0.85–1.16)	0.940
		Model 2	1.01 (0.90–1.13)	0.854	1.05 (0.90–1.23)	0.527	0.99 (0.85–1.17)	0.945
	Aggregated mental dimensions	Model 1	1.27 (1.14–1.43)	<0.001	1.22 (1.03–1.44)	0.022	1.23 (1.05–1.45)	0.012
		Model 2	1.22 (1.08–1.37)	0.001	1.18 (1.02–1.37)	0.029	1.18 (1.00–1.41)	0.056
**With abstainers as non-expose**	Aggregated physical dimensions	Model 1	1.02 (0.92–1.14)	0.691	1.06 (0.91–1.23)	0.461	1.01 (0.87–1.19)	0.867
		Model 2	1.02 (0.91–1.14)	0.730	1.06 (0.91–1.23)	0.489	1.00 (0.85–1.18)	0.964
	Aggregated mental dimensions	Model 1	1.30 (1.16–1.46)	<0.001	1.30 (1.10–1.53)	0.002	1.23 (1.04–1.45)	0.016
		Model 2	1.24 (1.10–1.40)	<0.001	1.25 (1.06–1.48)	0.009	1.16 (0.97–1.39)	0.100
**With those with comorbidities and abstainers**	Aggregated physical dimensions	Model 1	1.02 (0.92–1.14)	0.639	1.09 (0.95–1.27)	0.226	0.99 (0.85–1.15)	0.872
		Model 2	1.03 (0.92–1.14)	0.642	1.09 (0.94–1.27)	0.236	0.98 (0.84–1.14)	0.819
	Aggregated mental dimensions	Model 1	1.28 (1.15–1.43)	0.000	1.26 (1.07–1.47)	0.004	1.22 (1.04–1.43)	0.012
		Model 2	1.22 (1.09–1.37)	0.001	1.20 (1.02–1.41)	0.029	1.17 (0.99–1.39)	0.061
**Without binge-drinkers only on special occasions**	Aggregated physical dimensions	Model 1	0.96 (0.85–1.08)	0.529	1.00 (0.84–1.18)	0.963	0.95 (0.81–1.13)	0.593
		Model 2	0.96 (0.85–1.08)	0.465	0.99 (0.83–1.18)	0.929	0.94 (0.79–1.12)	0.485
	Aggregated mental dimensions	Model 1	1.32 (1.16–1.50)	0.000	1.26 (1.05–1.51)	0.014	1.30 (1.08–1.57)	0.005
		Model 2	1.26 (1.10–1.44)	0.001	1.20 (1.00–1.46)	0.055	1.25 (1.03–1.52)	0.022

Model 1: Adjusted for age, years of university studies, marital status, job occupation, being a health professional, physical activity, sleeping hours, alcohol intake, smoking status, total energy intake, adherence to the MDS, BMI, polypharmacy and prevalent diseases (diabetes, hypertension, hypercholesterolemia, arthritis, depression or pulmonary disease). Model 2: Model 1 also adjusted for the 4 years of follow-up quality of life data.

**Table 6 nutrients-15-01072-t006:** Multivariable adjusted * means of the different dimensions of quality of life as measured with the SF-36 according to binge-drinking (BD) habit after 8 years of follow-up and their 95% confidence intervals (CI).

	Multivariable-Adjusted* Mean (95% CI) for Non-Binge-Drinkers	Multivariable-Adjusted * Mean (95% CI) for Binge-Drinkers	*p*-Value
Aggregated physical dimensions	53.09 (52.94–53.24)	52.95 (52.73–53.17)	0.310
Physical functioning	94.77 (94.55–95.00)	94.38 (94.06–94.70)	0.062
Role physical	91.78 (91.17–92.38)	91.57 (90.70–92.44)	0.716
Bodily pain	78.87 (78.39–79.35)	78.12 (77.43–78.81)	0.093
General health	76.03 (75.70–76.36)	75.59 (75.11–76.06)	0.147
Aggregated mental dimensions	49.70 (49.50–49.91)	49.35 (49.05–49.64)	0.064
Vitality	66.32 (65.96–66.67)	65.78 (65.28–66.29)	0.105
Social functioning	91.71 (91.31–92.10)	90.94 (90.38–91.50)	0.035
Role emotional	89.68 (89.04–90.33)	89.26 (88.34–90.19)	0.485
Mental health	76.74 (76.44–77.05)	76.08 (75.65–76.52)	0.020

* Adjusted for age; years of university studies; marital status; job occupation; being a health professional; physical activity; sleeping hours; alcohol intake; smoking status; total energy intake; adherence to the MDS; BMI; polypharmacy; prevalent diseases (diabetes, hypertension, hypercholesterolemia, arthritis, depression or pulmonary disease); incidence of cancer, depression, diabetes or cardiovascular disease, and for the 4 years of follow-up quality of life data.

## Data Availability

The data presented in this study are available on request from the corresponding author.

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
