# Peer review of "Effect of Binge-Drinking on Quality of Life in the ‘Seguimiento Universidad de Navarra’ (SUN) Cohort"

_nutrients, 2023, doi:10.3390/nu15051072_

Round 1

Reviewer 1 Report

It is a very interesting and well conducted longitudinal study in which participants of SUN-cohort were followed for 8 years and the association between binge-drinking and quality of life was assessed. Binge drinking was associated to an increase in the probability of presenting worse physical functioning in the whole sample and women, and with worse score for the aggregated mental dimensions (whole sample and women), vitality (whole sample) and mental health (whole sample), even when adjusted for quality of life at 4 years of follow-up, and for alcohol intake (g/day), which means that this effect is independent of amount of alcohol consumed.  No substantial differences were found in stratified and sensibility analysis. But the association is different between those binge-drinkers that concentrate the alcohol consumption on a single occasion and those that spread the consumption throughout the week. It is highlighted that no positive effect was seen between the both variables, inspite of the perceived benefits of alcoholic beverages on well-being and mental health.

I have just some minor comments about the manuscript:

1) Methods 2.3. Once binge-drinking variable was measured as a number of alcoholic beverages on a weekday, on a weekend day and on celebrations and special occasions, how the average intake was obtained?

2) Results: What is the difference between figure 2 and the data presented in tables 2 and 3 (model 2)? Is it worth keeping the both (tables and figure)?

3) Discussion (line 303): I do not fully agree with the presented statement, since, although the magnitude of the OR found are higher in men, in general, significant associations (including after adjustments) were found for the general population or for women.

4) Limitations: I suggest to discuss that the same cutoff point was used for men and women, which can also underestimate the effect on the female population.

I congratulate the authors for the excellent manuscript presented.

Reviewer 2 Report

The manuscript presents the findings of a longitudinal analysis of the SUN cohort to examine the effects of binge drinking on quality of life 8 year later. The findings indicated that binge drinking was associated with poorer mental quality of life 8 years later, but not physical quality of life. Major strengths of the study are the longitudinal design with a very large sample. However, I have a number of major concerns about the study, detailed below.

1) In the introduction, it is not really clear what the current study contributes to the literature. The literature on binge drinking and quality of life is not reviewed; thus understanding how this study improves the existing literature is unclear. (The study contribution emerges in the discussion, but this should be clearly substantiated and reviewed in the introduction section.)

2) The premise for examining the effects of binge drinking on quality of life is not very well substantiated. In general, it appears that the authors are making the argument that binge drinking is generally a harmful behavior, but the mechanisms through which this behavior would lead to general quality of life differences seems vague and lacking distinction. Of course, extensive literature documenting the harms of binge drinking on many specific aspects of physical health (eg, metabolic functioning, etc) and mental health (eg, stress response, depression, etc) has been published. The authors also mention that many studies have examined more general quality of life. However, the premise of the study focusing on general quality of life is not well justified. For example, the four areas of the SF36 mental quality of life section focus on vitality, social functioning, emotional, and mental health; however there is no justification in the intro for focusing on any of these areas, or how they might be negatively impacted by binge drinking over time. 

3) To fully understand the role of binge drinking on quality of life, it seems like individuals who do not engage in binge drinking should also be included in analyses. 

4) Looking at high quality of life and lumping all others into a poorer quality of life category seems too simplistic; I would expect there to be substantial variability between those who have low and moderate quality of life.
